# Comparison of Hybrid Abdominal Flap versus Homodigital Neurovascular Island Flap for Fingertip Amputation

**DOI:** 10.3390/medicina58101483

**Published:** 2022-10-19

**Authors:** Yung Hoon Kim, Tae Hyung Kim, Jun Ho Park

**Affiliations:** 1Department of Plastic and Reconstructive Surgery, Seoul National University College of Medicine, SMG-SNU Boramae Medical Center, Seoul 07061, Korea; 2Department of Plastic and Reconstructive Surgery, Soonchunhyang University College of Medicine, Soonchunhyang University Gumi Hospital, Gumi 39371, Korea

**Keywords:** abdominal flap, fingertip amputation, bone graft

## Abstract

Amputation at the level of the distal phalanx is a common hand injury and is normally treated with replantation. However, if the level of injury is distal or the vessels of the stump have been crushed by injury, replantation cannot be a viable option. The aim of this study is to evaluate the functional and aesthetic outcomes of the “hybrid abdominal flap”, which consists of a random-pattern abdominal skin flap and an autologous bone graft. A retrospective analysis was performed on fingertip amputation patients from March 2019 to April 2021. The patients were managed by either hybrid abdominal flaps (HAFs) or homodigital neurovascular island flaps (HNIFs). The functional and aesthetic outcomes were assessed three months after the operations. In this study, a total of 20 fingers were treated with either HAFs or HNIFs. There was a significant difference in the range of motion (ROM) and the scar quality between the two groups. All of the flaps survived without flap necrosis, and non-union of the bone was not observed. The patients were satisfied with the appearance and function of their fingers, including the ROM and sensory aspects. Our novel HAF is a functionally and aesthetically valid option for fingertip amputations with crushed stumps.

## 1. Background and Objectives

Amputation at the level of the distal phalanx is a common hand injury, accounting for approximately 1% of all trauma-related injuries [1]. It is normally treated with replantation, revisional amputations, local flaps, and skin grafts. The basic goal of the treatment is to restore the function and the natural contour of the finger. Despite the advantages of revisional amputations for a quicker return to work while maintaining function, the shortening of the finger(s) after injury is aesthetically frustrating to patients [2]. Replantation of the fingertip amputation is also very difficult according to the level of the injury, even if the stump is clean-cut [3].

The homodigital neurovascular island flap (HNIF) is also a common option for fingertip amputation based on the concept of “like with like”, which utilizes a similar texture to the adjacent skin and soft tissues. It is a sensate flap that brings balanced tissue to reconstruct the tip of the finger. Normally, it is known to cover a relatively large area with an average maximal cutaneous advancement of 16 mm [4,5,6,7,8,9]. However, the limitations of pedicle length and traction on the pedicle often impose flexion of the proximal (PIP) and distal interphalangeal (DIP) joints, resulting in contracture and stiffness [10,11,12].

The random-pattern abdominal flap, first described by Biggs in 1898, could be a reliable and time-saving option to preserve a finger’s length and overall contour [13]. A successful sensory recovery after an abdominal flap coverage over a fingertip amputation has been proven [9]. However, donor site morbidity and limitation of motion until a flap division remain burdensome for the patient.

The aim of this study was to evaluate the functional and aesthetic outcomes of a random-pattern abdominal skin flap combined with a bone graft—the hybrid abdominal flap (HAF). Furthermore, the surgical outcomes were compared for the HAF and HNIF.

## 2. Materials and Methods

Our institutional review board (IRB No. 20-2022-75) approved this retrospective review of patients with immediate fingertip (Sebastin zone 1A and 1B) reconstruction performed from March 2019 to April 2021. The study was conducted in accordance with the Declaration of Helsinki and its later amendments. Informed consent was obtained from patients for all surgical procedures and wound management, as well as for the possible use of anonymized photographs. One group had undergone HAF (Group A), which combines the random-pattern abdominal flap with an autologous bone graft (Figure 1), and the other group had undergone HNIF (Group B). Informed consent was obtained after explaining the procedure to each patient. All procedures were performed in a standard manner by a senior plastic surgeon at a single institution (JH Park).

The indication for fingertip reconstruction was a patient whose distal stump was crushed or in ischemic status, and replantation was not affordable. Patients who underwent replantation, revision amputation, and skin graft were excluded. The patients were asked to choose either the HAF or the HNIF based on the differences in the results of the flaps and the postoperative management process.

The patients were followed up weekly for the first 4 weeks and, subsequently, at 3, 6, and 12 months after surgery. All complications were recorded and addressed accordingly. At the final follow-up, the range of motion (ROM), cold intolerance, scar assessment, sensation, and patient satisfaction were recorded. The active ROM for the injured fingers compared to the contralateral digits was assessed using a finger goniometer for the PIP and DIP joints. The sensation of the reconstructed and contralateral fingertips was assessed by measuring the static two-point discrimination (2-PD) using a DISKriminator device (Baseline 2-Point Discrima-Gon, New York, NY, USA). A scar assessment was performed using the Vancouver Scar Scale (VSS) by three blind observers who specialized in plastic surgery. The cold intolerance of the reconstructed and contralateral fingertips was recorded using the Cold Intolerance Symptom Severity (CISS) questionnaire. The patients’ aesthetic satisfaction was evaluated using the Aesthetic Numeric Analogue (ANA) score. The statistical analyses were performed using SPSS statistics for Windows, version 26.0 (IBM Corp., Armonk, NY, USA). Statistical significance was defined as *p* values of <0.05.

Surgical Technique of the Hybrid Abdominal Flap

The patients were placed in a supine position with the injured hand gently placed on an arm table or an arm board. A digital Allen test and an examination with a handheld doppler were routinely carried out to ensure the patency of the digital arteries in all patients.

The bony portion of the amputee was first analyzed. If an intact bone fragment from the stump could be utilized for the reconstruction, the fragment was rigidly fixated to the distal end of the recipient digit with interosseous wiring or a Kirschner wire. However, when the crushed stump or the intact bone were not available, an olecranon bone graft was performed. For the harvesting of the olecranon bone for a graft, a linear 3 cm incision was performed after palpating the anconeus and the flexor carpi ulnaris muscles. After elevating the periosteum, a sufficient autologous bone was harvested at 1.5 cm distal to the tip of the olecranon using an osteotome. The length and width of the harvested bone were analyzed based on the contralateral digit.

On the same side of the abdomen, a rectangular-shaped flap was designed along the territory of the inferior epigastric vessels in a relatively redundant area where it could provide comfort to the patient’s arm position. After raising the flap above the Scarpa’s fascia, the flap was wrapped with No. 5-0 Nylon around the injured part of the distal finger.

After the completion of the surgical procedure, a customized long-arm splint was used for flap stabilization. Elastic bandages and taping were employed to address the patient’s arm position for daily dressing and flap monitoring. All patients were encouraged to mobilize the shoulder and elbow joints. A division of the flap was performed after 3 weeks with local anesthesia.

Post-Division Management

During the division procedure, the condition of the flap was checked based on its marginal bleeding, temperature, and capillary refill. Any redundant portions of subcutaneous fat were removed to restore the natural contour of the fingertip. The Kirschner wire was simultaneously removed after identifying the graft status. Based on the flap condition, either a primary closure or negative-pressure wound therapy (NPWT) was performed. After confirming an improvement in the venous congestion, the flap was closed with No. 4-0 Nylon sutures.

Surgical Technique of the Homodigital Neurovascular Island FlapThe patients were placed in a supine position with the injured hand gently placed on an arm table or an arm board. The procedure was performed under tourniquet control. After measuring the defect size, a flap was drawn over either the distal ulnar or the radial aspect of the finger centered over the neurovascular bundle.A mid-lateral incision was then performed proximal to the outlined flap. The neurovascular bundle was dissected free of the surrounding tissues at the proximal phalangeal level. Once the neurovascular bundle was isolated, it was traced distally toward the base of the flap. Once the flap was confirmed directly over the pedicle, the central or volar aspect of the flap was then incised and dissected down to the flexor tendon sheath. The dissection proceeded under the flap in a central to lateral fashion, freeing the flap from the flexor tendon sheath. After the completion of the neurovascular bundle dissection, the flap was advanced and inserted into the area of deficit at the tip and sutured with fine absorbable sutures. A full-thickness skin graft was then harvested from the donor site of preference and used for coverage of the donor defect.After the completion of the surgical procedure, a customized short-arm splint was used for flap stabilization.

## 3. Results

The study cohort included 20 fingers of patients aged between 25 and 69 years (the mean age was 38.6 years). All patients suffered the injury at their workplace and presented on the day of the injury. The index (*n* = 11) and middle (*n* = 5) fingers accounted for the majority of the injuries. Eleven fingers were managed with HAFs, and nine fingers were managed with HNIFs. In the HAF group, the bony segment from the distal stump was utilized as an autologous bone graft in seven fingers, while four fingers were reconstructed with an autologous bone graft from the olecranon. The mean operative time was 43 min in patients who underwent the HAF and 38 min in patients with the HNIF. Out of the eleven patients with HAFs, seven patients were managed with an autologous bone graft from the distal stump and four patients underwent an autologous bone graft harvested from the olecranon.

The outcome data are summarized in Table 1. The active ROM in the DIP and PIP joints was better in the HAF group, with only the ROM of the DIP joint being significantly better (*p* = 0.031). The scar assessment (VSS) and patient satisfaction, which was determined using the ANA score, were significantly better in the HAF group (*p* = 0.012 and *p* = 0.003). The sensation at 12 months during the follow-up period—cold intolerance symptom severity (CISS)—showed no significant difference between the two groups. Complications such as flap necrosis, infection, and wound dehiscence were not observed in the study.

### 3.1. Case Reports

#### 3.1.1. Patient 1

A 25-year-old male patient experienced an amputation of the right second and third fingers by a heavy instrument. His third finger was amputated at the level below the lunula (Sebastin classification IB). An intact bony segment was harvested from the stump and fixated using interosseous wiring. An abdominal flap was wrapped around the defect. The satisfactory aesthetic and functional recovery were observed in the 6 months of follow-up (Figure 2).

#### 3.1.2. Patient 2

A 34-year-old female patient sustained a knife injury to the right second finger with a complete amputation below the lunula. An intact bony segment from the stump was autologously grafted to the recipient with a Kirschner wire (0.9 mm). The 5 cm-sized abdominal flap was wrapped around the defect. The patient had regained good contour and function at the 2-year follow-up without complications (Figure 3).

#### 3.1.3. Patient 3

A 50-year-old female patient sustained a knife injury to the right fourth finger with a complete amputation below the lunula. Considering the contralateral finger, a 1 cm × 0.3 cm-sized autologous bone was harvested from the olecranon and grafted to the recipient with a Kirschner wire (0.9 mm). The 4.5 cm-sized abdominal flap was wrapped around the defect. The patient had regained good contour and function at the 6-month follow-up (Figure 4).

#### 3.1.4. Patient 4

A 51-year-old male patient sustained an electrical saw injury to the right second finger with an amputation above the lunula. A 3.5 cm × 1.5 cm-sized, radially based HNIF was raised and advanced to cover the defect. Figure 5 shows the flap from the radial portion and its appearance 6 months after the flap coverage without complications.

## 4. Discussion

Fingertip amputation is a common hand injury, which is often accompanied by tendon or bone exposure. The objectives of treatment for fingertip amputations include the following: preservation of the best possible finger function with good sensation, maintenance of joint flexibility, and absence of cold intolerance and dysesthesias. Many surgical methods have been used to treat fingertip injuries. Revision amputations result in quick healing but with a shortened finger and a problematic scar. Flap coverage with local and distant flaps could be a good option when a stump is crushed. If the stump is viable, replantation would be considered the “gold standard” treatment [14,15]. When the vessels in the stump are too short, vein grafts can be performed to help anastomosis [16,17]. Although replantation is a commonly performed procedure, it is a very challenging surgery and is known to be almost impossible in the area above the lunula [3]. In addition, it is time-consuming work, requiring an intact stump. A full-thickness skin graft can cover the wound when there is no bone exposed, but the pulp will be insensitive.

In the case of a crushed stump, the HNIF is a common option for fingertip amputation based on the concept of “like with like”, which utilizes a similar texture of the adjacent skin and soft tissues [12]. It is performed in just one stage and does not involve any adjacent, uninjured digits. Foucher et al. have reported satisfactory results for patients treated with the HNIF technique. However, the main criticism of the HNIF is the potential for flexion and contracture of the PIP joint.

The random pattern abdominal flap is also the commonly used technique for fingertip amputations. Abdominal flaps have been considered random flaps since their first description in 1898 [13]. Various anatomical studies of the abdominal wall showed the pedicled abdominal flap based on the superficial inferior epigastric artery and superficial circumflex iliac artery [18,19,20]. A plane of flap elevation was chosen to be the supra-Scarpa’s fascia. The Scarpa’s fascia receives a robust blood supply from its epigastric vessels, which could create a good bone-healing environment. The literature supports long-term outcomes of successful pedicled flaps, which are equal to or even better than free flaps [21].

Pedicled flaps from the abdomen have several disadvantages, including patient discomfort due to immobilization of the entire hand for 2–3 weeks, the need for flap division, increased hospital stays, a bulky flap, and delayed hand therapy leading to stiffness [22]. Soft tissue coverage can also be achieved with an abdominal flap, although a shortening of the injured digital length is inevitable.

In this study, the authors prioritized the natural contour and function of the finger. For this, it is necessary not only to perform reconstructions of the soft tissues with a random-pattern abdominal flap, but also the overall length of the finger must be preserved. The length was preserved by a bone graft harvested from either a stump or the olecranon. This procedure is called the “hybrid abdominal flap”. Unlike the traditional abdominal flap, which is reconstructed only from soft tissues, the HAF is a reconstruction method that can restore both the bone and the soft tissues using the bony portion of the amputee or olecranon bone grafting. This can prevent bone loss, which can cause finger shortening, functional disability, and can be aesthetically frustrating to patients. Olecranon bone grafting was performed in the cases of stumps without a viable bone stump. It has been used in the treatment of non-unions of distal phalanx replantations [23]. Other options for autologous bone grafting are the iliac crest and the distal radius bone. The complications of bone harvesting from the iliac crest bone include abdominal hernia, neurogenic injuries, and vascular injuries [24]. The complications of harvesting bone grafts from the distal radius include tenosynovitis and neuroma of the superficial radial nerve [25]. However, an olecranon bone graft can be performed in the same surgical field, and its site is distant from important neurovascular structures. The harvesting procedure is simple and fast. However, it is not suitable for large defects and is not recommended for osteoporotic patients [26].

Compared to the HNIF group, which is a sensate flap, despite the concerns that the HAF group may have limited sensory recovery after surgery, there was no significant difference from the HNIF group when the sensation was actually measured using a static two-point discrimination. A sensory recovery has been confirmed clinically and histologically by the higher density of the nerves, which grew into the abdominal flap area [27]. With respect to appearance, the HAF group showed greater satisfaction than the HNIF group in the ANA score and the scar assessment (VSS). A full ROM of the distal interphalangeal joint was observed in all of the patients. In addition, the overall hand functions improved with time.

The simplicity and reliability of the HAF compared to replantation and other local and distant flaps have been proven in this study. Since its operation time is very short (mean time is 51 min) and there is a lower risk of morbidity compared to other procedures, the HAF can be performed under an axillary block or local anesthesia.

A hospital stay of more than 3 weeks for HAF is the main disadvantage. During the predivision period, although the patients’ fingers were fixed to their abdomen, they were encouraged to move their shoulders and elbow joints to prevent stiffness; consequently, none of the patients developed shoulder or elbow joint problems after the surgery.

This study has a few limitations. Firstly, although the study was conducted by one senior surgeon, the number of patients is small. Secondly, a longer follow-up is needed to analyze the important outcomes, such as contracture and scarring, in addition to long-term function of the injured finger. Lastly, a selection bias may occur by asking the patients which surgical procedure they prefer. However, given the preoperative explanations of the differences in postoperative care, as well as the results of the flaps (e.g., flap failure and scarring on finger), it can be stated that the patients received adequate information related to the surgery rather than being influenced by the surgeon’s preference.

## 5. Conclusions

In reconstructing fingertip amputations, if the stump is viable, replantation should be considered first. However, in many cases, the stump may become crushed or necrotic and cannot be used. In this case, the HAF is a good option that is safe and convenient. Despite the immobilization of the arm for three weeks, our novel HAF can preserve the length of the finger and provide reliable results with respect to functional, sensory, and aesthetic aspects.

## Figures and Tables

**Figure 1 medicina-58-01483-f001:**
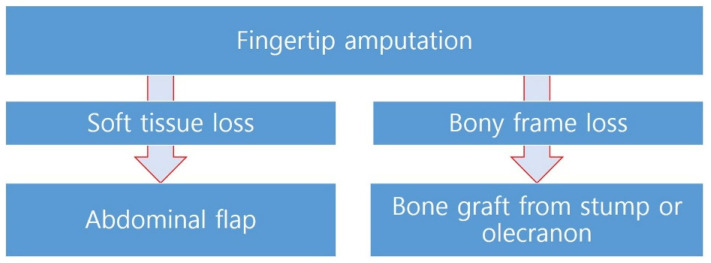
Algorithm for hybrid abdominal flap.

**Figure 2 medicina-58-01483-f002:**
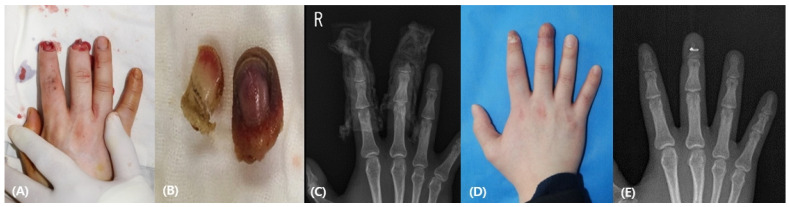
(**A**) A 25-year-old man was injured in his right distal phalanx of the second and third finger. The second finger was amputated above the lunula, while the third finger was amputated below the lunula. (**B**) The crushed stump was in an ischemic state with a necrotic color change. (**C**) Preoperative X-ray finding. (**D**) Postoperative photographic finding 6 months after the operation (**E**) Postoperative X-ray finding.

**Figure 3 medicina-58-01483-f003:**
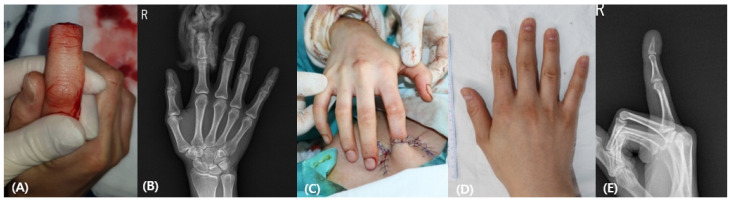
(**A**) A 25-year-old man was injured on his right distal phalanx of the second and third finger. The second finger was amputated above the lunula, while the third finger was amputated below the lunula. (**B**) The crushed stump was in an ischemic state with a necrotic color change. (**C**) Preoperative X-ray finding. (**D**) Postoperative photographic finding 6 months after the operation (**E**) Postoperative X-ray finding.

**Figure 4 medicina-58-01483-f004:**
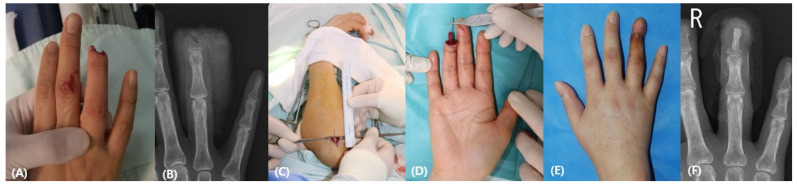
(**A**) A 50-year-old woman experienced a crush injury of the right fourth finger below the lunula. The stump was crushed without bony framework. (**B**) Preoperative X-ray finding. (**C**,**D**) The olecranon bone was harvested, and an autologous bone graft was performed with a k-wire. (**E**) Postoperative photographic finding 6 months after the operation (**F**) Postoperative X-ray finding.

**Figure 5 medicina-58-01483-f005:**
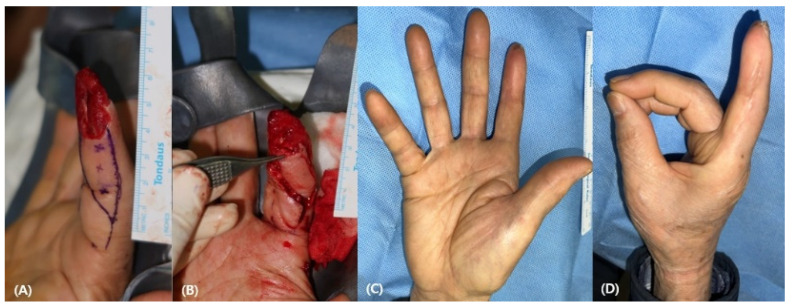
(**A**) 51-year-old man experienced a crush injury to the right second finger above the lunula. (**B**) A 3.5 cm × 1.5 cm-sized homodigital neurovascular island flap was elevated and transposed to the defect. (**C**,**D**) Postoperative photographic findings 6 months after the operation.

**Table 1 medicina-58-01483-t001:** Comparison of the results of both surgical methods.

		HAF Group	HNIF Group	*p* Value
Age of patients		39.2	37.8	0.794
Number of fingertips	Index	6	5	
	Middle	2	3	
	Ring	1	-	
	Little	2	1	
Donor site	Stump	7	-	
	Olecranon	4	-	
Operative time (min)		43	38	
Flap survival		11	9	0.803
ROM of DIP joint (degree)	Reconstructed	76.2 ± 4.8	70.3 ± 5.7	0.031
	Contralateral	77.8 ± 5.2	78.2 ± 6.1	-
ROM of PIP joint (degree)	Reconstructed	97.4 ± 3.4	94.7 ± 2.8	0.268
	Contralateral	98.2 ± 4.5	97.5 ± 5.2	-
Static 2-PD (mm)	Reconstructed	7.9 ± 1.3	7.3 ± 2.1	0.565
	Contralateral	3.6 ± 0.4	3.5 ± 0.5	
Scar assessment (VSS)		1.4	2.6	0.012
CISS score	Reconstructed	31.4	30.8	0.547
	Contralateral	11.6 ± 2.9	11.9 ± 1.8	0.982
ANA score		6.60	3.17	0.003

HAF: Hybrid abdominal flap. HNIF: Homodigital neurovascular island flap. ROM: Range of motion. DIP: Distal interphalngeal. PIP: Proximal interphalangeal. CISS: Convergence Insufficiency Sympotoms Surveys. ANA: Aesthetic Numeric Analogue.

## Data Availability

Not applicable.

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
