# Peer review of "Comparison of Hybrid Abdominal Flap versus Homodigital Neurovascular Island Flap for Fingertip Amputation"

_medicina, 2022, doi:10.3390/medicina58101483_

Round 1

Reviewer 1 Report

Thank you for a very interesting review regarding distal digital amputation flap management.

I have a few queries/comments: 

1. you state the patients were asked what flap they would like to receive (based on post-operative management). Doesn't this introduce a degree of bias to the selection process? I understand it may improve compliance, but certain patients will choose a process based on likelihood of success, and it may skew results. Unless you assume this bias is distributed across the whole (small) sample, it needs to be mentioned. Addionally, I would expect more patients to want the procedure that frees up their hand for movement, rather than keep a finger attached to the abdomen for 3 weeks.

2. the technique for HNIF isn't described (I assume it is performed in a standard fashion, but for completeness it should be mentioned)

3. the case studies show some good clinical points

4. the statistical analysis and data points ar sound.

Author Response

Dear Editors & reviewers,

Thank you for your handling the review of our manuscript. We were pleased to see the reviewers’ numerous valuable comments on the study. We have carefully incorporated all of the reviewers’ suggestions into the revised manuscript. We hope our revision could cover all your comments sufficiently.

Reviewer

Comment 1. you state the patients were asked what flap they would like to receive (based on post-operative management). Doesn't this introduce a degree of bias to the selection process? I understand it may improve compliance, but certain patients will choose a process based on likelihood of success, and it may skew results. Unless you assume this bias is distributed across the whole (small) sample, it needs to be mentioned. Addionally, I would expect more patients to want the procedure that frees up their hand for movement, rather than keep a finger attached to the abdomen for 3 weeks.

Response: We appreciate your candid comments. We replaced the sentence at Materials and Methods in line 63-66 in our revised manuscript as follows;

Patients who underwent replantation, revision amputation, and skin graft were excluded. Patients were asked to choose either HAF or HNIF based on the differences in results of flaps and postoperative management process

Also, we added limitation in the discussion section in line 267-271 in our revised manuscript as follows;

Lastly, selection bias may occur by asking patients which surgical procedure they prefer. However, given the preoperative explanations of differences in postoperative care as well as results of flaps (ex. flap failure, scar on finger) it can be regarded as patients received adequate information related to surgery rather than influenced by the surgeon’s preference.

Comment 2. the technique for HNIF isn't described (I assume it is performed in a standard fashion, but for completeness it should be mentioned)

Response: Thank you for your comments. We have add the sentence about technique of NHIF in line 112-128 in our revised manuscript as follows ;

The patient was placed in a supine position with the injured hand gently placed on an arm table or arm board. The procedure is performed under tourniquet control. After measuring the defect size, a flap is drawn over either the distal ulnar or radial aspect of the finger centered over the neurovascular bundle.

A mid-lateral incision is then made proximal to the outlined flap. The neurovascular bundle is dissected free of surrounding tissue at the proximal phalangeal level. Once the neurovascular bundle has been isolated, it is traced distally toward the base of the flap. Once the flap is confirmed directly over the pedicle, the central/volar aspect of the flap is then incised and dissected down to the flexor tendon sheath. Dissection proceeds under the flap in a central to lateral fashion, freeing the flap off the flexor tendon sheath. After completion of the neurovascular bundle dissection, the flap is advanced and inserted into the area of the deficit at the tip and sutured with fine absorbable sutures. A full-thickness skin graft is then harvested from donor site of preference and used for coverage of the donor defect.
 After the completion of the surgical procedure, a customized short arm splint was used for flap stabilization.

Comment 3. the case studies show some gooqd clinical points

Comment 4. the statistical analysis and data points are sound.

Response: Thank you for your comments.

Reviewer 2 Report

The content is very interesting and suitably presented. However, the result section contained only 4 case studies. Also, the discussion is very poor. I suggest improving the result and discussion part, and incorporating more discussion dimensions. The author should try to include and discuss patient feedback, the cost-effectiveness of both methods, the number of clinical visits, duration of stay in the hospital, and pain management. The article should contribute positively to the research group. Till now the article doesn't have such information or a conclusion.

Author Response

Response to Reviewers

" Comparison of Hybrid Abdominal Flap Versus Homodigital Neurovascular Island Flap for Fingertip Amputation"

Dear Editors & reviewers,

Thank you for your handling the review of our manuscript. We were pleased to see the reviewers’ numerous valuable comments on the study. We have carefully incorporated all of the reviewers’ suggestions into the revised manuscript. We hope our revision could cover all your comments sufficiently.

Reviewer

The content is very interesting and suitably presented. However, the result section contained only 4 case studies. Also, the discussion is very poor. I suggest improving the result and discussion part, and incorporating more discussion dimensions. The author should try to include and discuss patient feedback, the cost-effectiveness of both methods, the number of clinical visits, duration of stay in the hospital, and pain management. The article should contribute positively to the research group. Till now the article doesn't have such information or a conclusion.

Response: We appreciate your candid comments. The statistics in the results section included all the results of the entire patient group (20 cases) to derive the results, and only 4 cases that we wanted to introduce among them were shown as examples. It does not include only 4 cases in the Result section.

And, As can be seen in our manuscript “the HAF is a reconstruction method that can restore both bone and soft tissue using the bony portion of the amputee or olecranon bone grafting. This can prevent bone loss, which can cause finger shortening, functional disability, and can be aesthetically frustrating to patients.”, the advantages and effectiveness of HAF (ex. Prevent bone loss) were described in the discussion section.

Also, looking at “Compared to the HNIF group, which is a sensate flap, despite concerns that the HAF group may have limited sensory recovery after surgery, there was no significant difference from the HNIF group when sensation was actually measured using static two-point discrimination. Sensory recovery had been confirmed clinically and histologically by the higher density of nerves, which grew into the abdominal flap area [27]. With respect to the appearance, the HAF group showed greater satisfaction than the HNIF group in ANA score and scar assessment (VSS). Full ROM of the distal interphalangeal joint was observed in all the patients. In addition, overall hand function improved with time. The simplicity and reliability of the HAF compared to replantation and other local and distant flaps have been proven in this study. Since its operation time is very short (mean 51 minutes) and there is a lower risk of morbidity compared to other procedures, the HAF can be performed under an axillary block or local anesthesia.”, we compared with HNIF at cosmetic, sensory and functional aspects, and proved that HAF can bring more satisfactory results to patients based on its simplicity and safety.

Round 2

Reviewer 2 Report

The authors have incorporated all the suggestions satisfactorily. The manuscript can be considered for publication.